# Achieving Control of Asthma in Children in Africa (ACACIA): protocol of an observational study of children's lung health in six sub-Saharan African countries

Gioia Mosler [1], Victoria Oyenuga,[1] Emmanuel Addo-Yobo,[2] Olayinka Olufunke Adeyeye,[3] Refiloe Masekela,[4] Hilda Angela Mujuru,[5] Rebecca Nantanda,[6] Sarah Rylance,[7] Ismail Ticklay,[5] Jonathan Grigg[1]

For numbered affiliations see end of article.

**Correspondence to**
Dr Gioia Mosler;
g.mosler@qmul.ac.uk

## ABSTRACT

**Introduction** Little is known about asthma control in the rising number of African children who suffer from this condition. The Achieving Control of Asthma in Children in Africa (ACACIA) study is an observational study collecting evidence about paediatric asthma in urban areas of Ghana, Malawi, Nigeria, South Africa, Uganda and Zimbabwe. The primary objectives are: (1) to identify 3000 children aged between 12 years and 14 years with asthma symptoms; and (2) to assess their asthma control, current treatment, knowledge of and attitudes to asthma and barriers to achieving good control. Secondary objective is to develop interventions addressing identified barriers to good symptom control.

**Methods and analysis** Each centre will undertake screening to identify 500 school children with asthma symptoms using questions from the Global Asthma Network's questionnaire. Children identified to have asthma symptoms will fill in a digital survey, including: Asthma Control Test, questions on medication usage and adherence, medical care, the Brief-Illness Perception questionnaire and environmental factors. Exhaled nitric oxide testing and prebronchodilator and postbronchodilator spirometry will be performed. A subgroup of children will participate in focus group discussions. Results will be analysed using descriptive statistics and comparative analysis. Informed by these results, we will assess the feasibility of potential interventions, including the adaption of a UK-based theatre performance about asthma attitudes and digital solutions to improve asthma management.

**Ethics and dissemination** The ACACIA study has been reviewed by the Queen Mary University of London Ethics of Research Committee in the UK. All African centres have received local ethical approval for this study. Study results will be published in academic journals and at conferences. Study outputs will be communicated to the public via newsfeeds on the ACACIA website and Twitter, and through news media outlets and other local dissemination.

**Trial registration number** 269211.

## Strengths and limitations of this study

► The Achieving Control of Asthma in Children in Africa (ACACIA) study will record the number of children with asthma symptoms and proportion of these children with severe symptoms through school-based screening in Ghana, Malawi, Nigeria, South Africa, Uganda and Zimbabwe.

► The ACACIA survey and focus group results will provide a wide range of information about barriers faced by children with asthma symptoms that prevent good asthma control and management.

► Although the study will recruit from six urban areas in sub-Saharan African countries, it can be readily expanded to other areas in other countries.

► ACACIA plans to test potential asthma management interventions with a special focus on asthma symptom control and hopes to obtain further funding for full intervention development in a second stage, which would address barriers to good asthma management identified through the study.

► One limitation is that children will be recruited in urban areas only, and previous studies have shown that asthma control levels in urban and rural areas are different with more severe asthma in urban compared with rural areas.

## INTRODUCTION

Until recently, childhood asthma in Africa was not considered to be a significant health issue. Surveys of African children (age 13–14 years) in schools have found that the proportion of children with symptoms of asthma ranges between 10% and 20% in Central Africa, increasing to over 20% in South Africa. This high prevalence of asthma symptoms among children in Africa is thought to be partly due to the increase in urbanisation in Africa, which are linked to changes in lifestyle and environmental risk factors.[1] Uncontrolled

asthma is associated with missed school days, repeated hospitalisation, risk of airway remodelling and general poor quality of life in the affected children.[2]

There have been few studies on childhood asthma in Africa, most of which focus on symptom prevalence or associations of asthma prevalence with allergies or air pollution.[1–4] Little is known, however, about the impact asthma has on the life of young people and children in sub-Saharan African countries. A report from the Global Asthma Network (GAN) speculates that asthma control in African children is poor. Reasons for poor control are suspected to lie in the affordability and availability of asthma inhalers and in a lack of diagnosis, poorly coordinated asthma management, a lack of asthma knowledge in children, teachers and parents and stigma associated with this long-term condition.[2] One study by Iraqi and Mahraoui[5] assessed influences of asthma control in 521 children across seven African countries, of which Senegal, Niger and Mali were located in sub-Saharan Africa. The study found that 45% of the children in the study had suboptimal asthma control according to the Global INitiative for Asthma (GINA) questions devised to assesses asthma control. The study, furthermore, found that several factors were associated with poor control, including easy access to healthcare and failure to carry out patient education. One study from Tanzania reports about fears of young people without asthma to play, sleep or eat with their peers who have asthma, indicating an existing stigma about asthma.[6] There is a need for a comprehensive assessment of asthma control, asthma symptoms and lung function in school children. The identification of possible facilitators and barriers for good symptom control will inform the design of relevant and scalable interventions.

The Achieving Control of Asthma in Children in Africa (ACACIA) study will draw on the methodology of the UK School-based Asthma Project (SAP). The SAP found that 46% of 766 London (UK) children and young people (aged 12–18 years) had suboptimal asthma control. Further results from survey and focus group data revealed that many of them face a range of barriers to good asthma management, including lack of knowledge, forgetfulness and perceived stigma around asthma.[7] Based on this evidence, an intervention, called the 'My Asthma in School Programme', was developed, comprising two components: (A) a theatre workshop for all children in school year groups 7 and 8 and (B) a self-management workshops for children with asthma. A theory-based cluster randomised controlled trial is currently running in London to test this intervention with 3-month, 6-month and 12-month follow-up.[8]

The ACACIA study will collect observational data about childhood asthma in a similar manner as the SAP in order to build the evidence base for targeted and effective interventions in the future. ACACIA will collect data related to asthma from adolescents aged 12–14 years across sub-Saharan Africa. The study wants to understand asthma during adolescence, when there is a shift from parental to personal symptom management and when optimal self-management standards of asthma could be introduced in a future intervention. ACACIA will furthermore focus on urban areas in sub-Saharan Africa, as prevalence of childhood asthma is particularly high in urban areas of the region.[1]

## Study aims and objectives

The ACACIA study aims to improve understanding of the burden that asthma imposes on school children, including the proportion of children with symptoms, symptom severity and asthma control. A secondary aim of the study is to deliver evidence of the highest quality on the barriers to achieving good asthma control that contributes to poor asthma outcomes in urban African children. The primary objectives of this observational study are to: (1) identify a total of 3000 children between 12 years and 14 years of age with asthma symptoms from Ghana, Malawi, Nigeria, South Africa, Uganda and Zimbabwe and (2) to assess their asthma control, lung function, current treatment and access to care, knowledge of and attitudes to asthma, as well as to identify the barriers to achieving good asthma control. The secondary objective is to explore ways to address the identified barriers, including the adaption of the '*In Control*' theatre performance about understanding of asthma, from the SAP.

## METHODS AND ANALYSIS

The main study will follow a cross-sectional observational design. The study comprises four stages:

Stage 1: screening for asthma symptoms and asthma diagnosis in a Breathing Survey in school using questions from the GAN questionnaire.[9] Participants, aged between 12 years and 14 years, in school will be recruited, as presented in the participant flow chart (figure 1). Children identified during screening with either wheezing symptoms in the previous 12 months, or those who report that they ever had asthma, or both will be invited to take part in the core data collection (stage 2).

Stage 2: core data collection at school includes the ACACIA questionnaire, prebronchodilator and postbronchodilator spirometry and exhaled breath nitric oxide (FeNO) measurements.

Stage 3: participants with symptoms of asthma will be invited to take part in focus group discussions on the conclusion of core data collection.

Stage 4: potential interventions will be developed based on intermediate results of the data collection. One specific intervention, an educational theatre performance, will be adapted from a UK-based play about asthma, called '*In Control*', piloted and tested for its effectiveness and impact on understanding asthma by young people in Africa.

Informed parental/guardian consent and child assent is collected in accordance with both country/local regulations and UK ethics regulations at three stages during data collection: (1) screening survey, (2) core data collection and (3) focus groups (see also figure 1) in five of the countries. In one country (Uganda), consent and assent for stages 1 and 2 are collected together, as this was the preferred design in regards to the country's ethical approval.

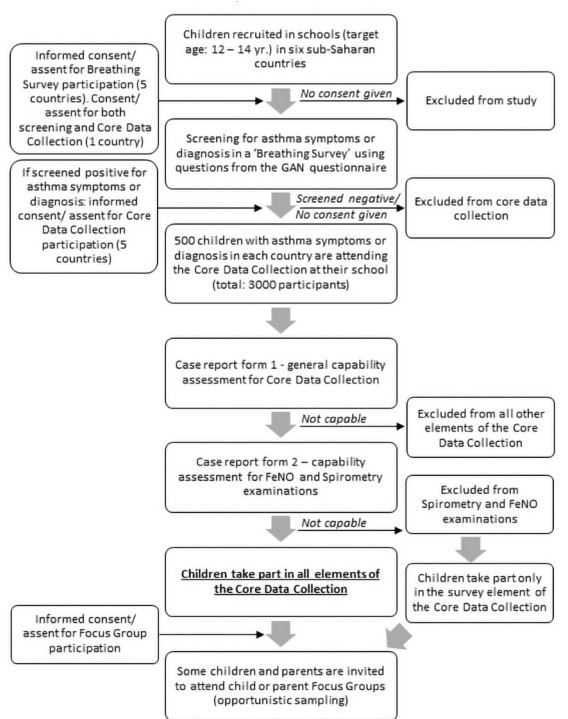

**Figure 1** Participant flow chart. FeNO, exhaled nitric oxide; GAN, Global Asthma Network.

The study will be conducted simultaneously in schools of six African urban areas: Kumasi, Ghana; Blantyre, Malawi; Lagos, Nigeria; Durban, South Africa; Makerere, Uganda; and Harare, Zimbabwe. Work in each of these urban areas is led by a principle investigator at a local university and conducted by their team of researchers and fieldworkers.

## Preparing for data collection

To prepare for data collection, protocols and guidelines were developed centrally, as well as a digital platform for data entry. The digital platform comprises a central research database, hosted by Microsoft's Azure Platform in the UK, which receives and stores the anonymised data collected by all field teams, and local databases in each centre, stored on servers within the university, which allow storage of personal data. A suite of bespoke software enables for data transfer and reporting. In the African study centres, research and fieldwork teams are recruited and trained. Field workers will be trained to perform quality assured spirometry using the NDD Easy On-PC and the use of the NIOX VERO to measure Fractional exhaled Nitric Oxide (FeNO). FeNO and spirometry training is in accordance with American Thoracic Society (ATS) and European Respiratory Society (ERS) standards 2005. Testing will be carried out by the field workers, who have undergone training, using the NDD Easy On-PC and the NIOX VERO. Standard operating procedures for spirometry and FeNO testing will also be provided to ensure standardisation and quality. Data entry and local storage systems are implemented with remote support from the central data management team.

All study materials, including information sheets and consent forms for parents, as well as assent and questionnaires for children are translated into local languages in areas where English is not universally used: Chichewa in Malawi (Blantyre), IsiZulu in South Africa (Durban), Luganda in Uganda (Makerere) and Shona in Zimbabwe (Harare). Translations are using forwards and backwards translations. For the Asthma Control Test (ACT; licence donated to Queen Mary University of London (QMUL) by GSK) translations are validated by the Mapi Research Trust.

## Recruitment

Each African study centre will recruit schools to the project until they have identified 500 children using a short screening questionnaire, called Breathing Survey. Recruitment of schools to the project started in July 2019, and school-based data collection is planned to finish before January 2021. The target age for the central analysis is 12–14 years. In some centres, school classes contain a wide age range of pupils as children progress through the school system at different rates, dependent on a number of factors, including academic ability, and periods of school absence. It was therefore decided that centres can include older children of 15 and 16 years to improve ease of access to children within schools. This additionally collected data will be excluded from the planned central analysis. Local research centres may decide to use the data in an analysis at a later stage. During school selection, researchers aim to reflect country-specific proportions of children educated in different school types (eg, private schools and government-run schools) and aim to spread selected schools across the urban region in order to reflect geographical differences in socioeconomic status. The number of schools recruited by each centre depends on the size of schools, as well as on asthma prevalence. For areas with low school attendance, additional recruitment of children attending acute care was considered. Subsequent feedback from local principal investigators confirmed however that only a small part of children would not attend school. Recruitment in schools was therefore considered to be sufficient.

## Screening for asthma symptoms

Prior to completing the screening survey, informed consent and assent will be obtained in accordance with local and UK ethical review board requirements. The research teams are working closely with the schools to obtain consent from the parents ahead of data collection. Information sheets about the study are provided to parents and children, outlining the research, the participant's involvement and important aspects of participation, such as the right to withdraw. The local research teams are available for any further questions and for assistance with reading of the information sheet, if necessary. Researchers will visit school classes containing children of the target age group to facilitate the filling in of the Breathing Survey, provided consent and assent is given. The questions used in the Breathing Survey are taken

from the validated GAN questionnaire to screen for asthma symptoms, using its demographic questions and questions 2–7 about breathing.[9]

## ACACIA survey and lung function tests

In a second stage, the adapted online survey will be delivered to children in schools by the research team who identified as having asthma symptoms or asthma diagnosis in stage 1. Following existing studies, the percentage of children with asthma symptoms is expected to be between 10% and 20%.[1] In addition, children with asthma symptoms will undergo spirometry and FeNO measurement.

The ACACIA survey is adapted from a version of a UK-based SAP survey[7]; a non-digital list of the survey questions can be found in the online supplementary file. In schools where broadband is not available, the survey can be filled in offline; answers will be uploaded when connectivity is restored. The usage of electronic devices with long battery lives is encouraged in case of power cuts. Students who have difficulty using the digital survey will be supported by the research team.

The ACACIA survey includes:
1. Personal details.
2. ACT and GINA questions to assess asthma control.
3. Medication and adherence.
4. Medical attention and access to care.
5. School attendance and attainment.
6. Brief Illness Perception Questionnaire – Asthma.
7. Asthma-related knowledge.
8. Smoking, home living and exposure.
9. Home location and distance to nearest major road (researcher assisted).
10. School specifications: playground surface, distance to next major road (researcher).

Several sections of the survey will be using questions successfully applied in the SAP survey,[7] including: (A) medication adherence, (B) unscheduled care, (C) illness perception of asthma (measured using the Brief Illness Perception Questionnaire),[10] (D) school absences, (E) asthma knowledge and (F) smoking and parental smoking. Additional questions about access to healthcare and about environmental exposure were adapted from the set of International Multidisciplinary Programme to Address Lung health and TB in Africa (IMPALA) questionnaires.[11]

Asthma control is one of the primary outcomes of this study and will be assessed using the ACT,[12] as well as GINA assessment of symptom control and future risk.[13] The ACT is a validated tool for assessing asthma control in people from the age of 12 years. The ACT consists of five items that assess activity limitation, shortness of breath, nighttime symptoms, use of reliever medication and overall rating of asthma control over the previous 4 weeks. The test score gives an indication as to whether the subject has controlled asthma or uncontrolled asthma. GINA uses four questions about symptoms, reliever usage and activity limitations to assess if a participant has well-controlled, partly controlled or uncontrolled asthma symptoms with a set of four questions.

Further insights into asthma symptoms will be provided by lung function data collected using spirometry and FeNO testing. Case report forms will be used to gather information on any medications a participant is taking and the presence of any absolute or relative contraindications to FeNO or spirometry measurement. Calibration verification (linearity check) will be performed every day, prior to testing, at each data collection location. Prebronchodilator and postbronchodilator spirometry will be performed for all participants who have no contraindications. Four hundred micrograms of salbutamol will be given, postbaseline spirometry, via a pressurised Metered-Dose Inhaler (pMDI) and spacer. Spirometry quality will then be reviewed and agreed by two spirometry specialists before deemed acceptable for use in analysis. The reference values used will be calculated from the global lung function (GLI) 2012 equations.[14]

After data have been collected at the school, children identified with asthma symptoms without a doctor's diagnosis of asthma, will receive a letter to the parents advising them to seek appropriate local healthcare or will be invited directly by local clinics where possible.

## Focus groups

After the core data collection, focus groups will be held to deepen the research team's understanding of possible facilitators and barriers to effective asthma management and to discuss possible components for a school-based asthma intervention. At each centre, ACACIA participants from at least three schools are selected to take part in 1-hour long focus group discussions with five to eight participants. In each school, there will be: (A) one discussion with children who have an asthma diagnosis and (B) one discussion with children who have symptoms of asthma but no diagnosis. An additional six focus groups per centre will be held with parents and teachers. Informed consent and assent is collected for the focus group attendance in accordance with local and UK ethics regulations. Based on preliminary findings from the collected questionnaire data, several focus group questions will be developed for all study centres; each centre can then add topics of specific interest to their local context to the focus group discussions. These focus groups will be held by trained local researchers in each country. Focus group conversations will be recorded. A detailed protocol for the facilitation and analysis of focus groups will be developed for all centres.

## Outcomes and analysis

The primary outcome from this study will be the identification of the proportion of children with asthma symptoms and the proportion of those who have poor asthma symptom control in children aged 12–14 years in six sub-Saharan African countries. Control will primarily assessed by using:

- ► Percentage of children with asthma symptoms or asthma diagnosis based on questions from the GAN screening tool.
- ► ACT: using validated ACT. The minimum score is 5 (poor control of asthma), and the maximum score is 25 (complete control of asthma). An ACT score >19 indicates well-controlled asthma.
  Alternative ways of assessing asthma control will be explored, including[15]:
- ► Asthma control according to GINA: GINA questionnaire using four questions, assessing control of asthma symptoms. Outcome is 'well controlled' if none of the four questions is answered 'Yes', 'partly controlled' if one or two of the four questions is answered 'Yes', uncontrolled, if three or four of the four questions is answered 'Yes'.
- ► FeNO measurement: a significant FeNO measurement will be deemed as a reading of ≥35 ppb, as per ATS/ERS guidelines for children.[16]
- ► Spirometry: spirometry will be classified as obstructed if the forced expiratory volume in 1 s ($FEV_1$)/forced vital capacity (FVC) ratio is less than the lower limit of normal (LLN). Obstructed spirometry will undergo a grading of severity based on the prebronchodilator $FEV_1$ z-score. Postbronchodilator reversibility will be defined as an increase in the highest $FEV_1$ of ≥12% and 200 mL from baseline.[17]
  Spirometry parameters used for analysis include:
  - FEV1, in litres, and expressed as percent predicted and z-score.
  - FVC, in litres, in per cent predicted and z-score.
  - $FEV_1$/FVC ratio, $FEV_1$/FVC LLN.
  - Forced Expiratory Flow (FEF) 25-75 in litre/second and in per cent predicted.
  Secondary outcomes:
- ► Assessment of barriers to good control:
  - Current treatment of asthma: set of questions about current medication, reported using descriptive statistics, and percentages
  - Adherence to medication: set of questions about adherence to medication, reported as descriptive statistics and percentages, as well as free-text comments.
  - Access to medical care for asthma: set of questions related to access to medical care, reported using descriptive statistics and percentages.
  - Understanding of asthma: set of questions about asthma knowledge, maximum knowledge score is 13, minimum knowledge score is 0 and total range is 13. Higher values represent better knowledge.
  - Illness perception: the Brief Illness Perception Questionnaire for asthma. Each of the eight items in the Brief Illness Perception Questionnaire has a minimum score of 0 and a maximum score of 10. Overall score that represents the degree to which the illness is perceived as threatening or benign.[10 18]

- ► Asthma-related time off school: set of questions asking about asthma-related time off school, reported as descriptive statistics and percentages.
- ► Assessment of environmental factors:
  - Smoking: a set of questions assessing active and passive smoking, reported as descriptive statistics and percentages.
  - Environmental factor assessment: a set of questions related to the environment of young people with asthma symptoms, reported using descriptive statistics and percentages.

Correlations with measures of asthma control to other variables, such as adherence to medication, and illness perception will be explored. Cross-analyses and comparison of the data between centres will also be undertaken. A detailed statistical analysis plan is being developed with the assistance of statistician.

Focus group discussions will be transcribed and keywords identified. Qualitative results from the focus groups will in addition inform the direction and acceptability of interventions aimed at removing barriers to poor asthma control.

Transcripts of the focus groups will be analysed using qualitative methods. After familiarising themselves with the data, a minimum of two researchers will use an inductive coding process, and thematic analysis will be used to determine patterns in the responses. Themes across data from the various focus group discussions will be identified and coded by each individual researcher manually. Consensus will be reached on the major themes identified using an iterative process. NVivo 11 is used to assist this component of the analysis and to enter transcribed data within codes, categories and themes.

### Intervention development

A preliminary analysis of the data from the survey and focus groups will inform ideas for possible interventions aimed at improving asthma outcomes and understanding around asthma in African children. One component of an intervention that will be assessed is a UK theatre intervention (play) adapted for an African context. The ACACIA study also aims to improve the digital resources available for asthma-related services and for children who suffer from asthma. Digital solutions aimed at improved asthma self-management, such as smart peak flow metres and wheeze detectors, which can send information about symptoms to smart phone apps, will be tested by some of the study participants. Transferability of data infrastructure built for ACACIA study to other studies or applications will also be explored.

### Theatre intervention development and pilot

The original 'In Control' theatre was written by the Nigerian-born playwright Tunde Euba in collaboration with The Tramshed – Greenwich and Lewisham Young People's Theatre as an educational performance, aimed at changing attitudes and understanding around asthma. The play revolves around a young protagonist with asthma

in a school setting. It is described in a *Lancet* perspective article.[19] In a UK-based feasibility study, 98.7% of 1798 young people in the audience said '*In Control*' was enjoyable and 84.6% said that the performance at least somewhat changed how they think or feel about asthma.[20] The theatre will be adapted to the cultural context of sub-Saharan Africa by the original play writer in close collaboration with local arts and theatre groups, and local actors will be trained to perform the play. A pilot study will explore its potential to change understanding of asthma. At each study centre, the pilot study will involve performances in at a minimum of five schools with two performances in each school to audiences of up to 120 children (target age 12–16 years) and five additional performances to local community groups, which will be defined after initial community contacts during project recruitment. Outcomes will be measured using short before and after questionnaires about knowledge and attitude towards asthma with additional questions on acceptability in the after-performance questionnaire. Results will be summarised using descriptive statistics and inform any potential future development of a full intervention, if further funding is provided.

### Public and patient involvement

Children of the target age group have been involved during the study development and, for example, commented on the ACACIA questionnaire and the theatre adaptation. Beyond that, all study sites go through a guided process of developing their Public and Patient Involvement and Engagement (PPIE) activities. PPIE guidelines have been developed centrally in collaboration with the public engagement team at QMUL. A PPIE lead at each study site works with the central team to develop a PPIE plan, specific to their local context, and in accordance with the study's aims and objectives. Wherever possible, PPIE activities will be evaluated using established toolkits.

## ETHICS AND DISSEMINATION
### Ethical review

The ACACIA study has been reviewed by the Queen Mary Ethics of Research Committee in the UK. Each collaborating African centre received ethical approval for the study locally with: Malawi: The College of Medicine Research Ethics Committee, Reference No: P.10/18/2494, University of Malawi. Nigeria: Lagos State/Lagos State University Teaching Hospital Health Research and Ethics Committee (NHREC04/04/2008), Reference No: LREC 06/10/1084. Uganda: Mulago Hospital Research Ethics Committee (MHREC) number MHREC 1514; Institutional Review Board, and National Council for Science and Technology number SS 4940. South Africa: Biomedical Research Ethics Committee of University of KwaZulu Natal BREC number: BF002/19 and Department of Basic Education KwaZulu Natal and Department of Health KwaZulu Natal Zimbabwe: The City Health Ethics Board, Joint Research Ethics Committee, the Medical Research Council of Zimbabwe and the Research Council of Zimbabwe Ghana: Committee on Human Research, Publication and Ethics a joint committee of the School of Medical Science, Kwame Nkrumah University of Science and Technology and the Komfo Anokye Teaching Hospital, which is registered with the US Department of Health and Human Services, Office for Human Research Protection Study direction and progress will be guided by an international ACACIA steering group, chaired by Heather Zar.

### Dissemination

Study results will be published in national and international peer-reviewed journals and at conferences. Scientific results will be published as soon as it is reasonably practical and in compliance with the terms of the funding body. Publication will only be made after QMUL has received written permission from the funding body. Authorship will not necessarily be restricted to individuals named on this protocol; neither is authorship guaranteed to individual named on this protocol. Contributors who do not meet authorship criteria will be listed in 'Acknowledgements'.

General study information will be displayed on the study's website, as well as through its Twitter account. Furthermore, dissemination of study results in the community is planned as part of PPIE events and activities.

### Data sharing, access and release

The ACACIA study collects data from children in six African countries. Data will be collected by local research teams in each of the African centres. Participant's identifiable data will be pseudoanonymised before data are analysed or shared with the other centres through a digital platform system. ACACIA study will release curated and coded data for open access after the results have been published, within the limits of relevant data protection acts.

### Data protection and confidentiality

Data collected from participants are stored locally in accordance with local and UK data protection legislations. The research team will be mindful to implement the highest data protection standards in all countries, especially where little or no local legislation exists. Access to identifiable data will only be possible within each country with local data access restricted to a limited number of researchers. The ACACIA central platform is hosted behind firewalls within a UK data storage facility. Storage and destruction of data are compliant with the UK Data Protection Act 2018. Only limited potentially identifiable data are contained within the central platform, specifically:

1. Age.
2. Ethnicity.
3. Gender.
4. School (changed into an identifier).
5. Name (changed to an identifier).
6. Exact home location (changed into a square area (Mercator projection, reduced to two decimals).

## Data monitoring

Intermittent random audit of data quality will be performed by members of each local investigating team. Audit by members from the coordinating centre in the UK will additionally be performed, including spot checks of entered data, site visits during data collection and equipment checks.

**Author affiliations**
[1]Centre for Genomics and Child Health, Blizard Institute, Barts and The London School of Medicine and Dentistry, Queen Mary University of London, London, UK
[2]School of Medicine and Dentistry, College of Health Sciences, Kwame Nkrumah University of Science and Technology, Kumasi, Ghana
[3]College of Medicine, Lagos State University, Ojo, Nigeria
[4]Paediatrics and Child Health, Nelson R Mandela School of Clinical Medicine, University of KwaZulu-Natal, Durban, South Africa
[5]College of Health Sciences, University of Zimbabwe, Harare, Zimbabwe
[6]Lung Institute, Makerere College of Health Sciences, Makerere University, Kampala, Uganda
[7]Malawi-Liverpool -Wellcome (MLW) Trust Clinical Research Programme, Wellcome Trust, Blantyre, Malawi

**Acknowledgements** The authors would like to thank Professor Farida Fortune from Queen Mary University of London and her technological development team with Andy Carter and Ian Sempill for their hard work towards developing innovative technological solutions for the study. Further thanks go to Jeremy James, Director of The Tramshed – Greenwich and Lewisham Young People's Theatre, who collaborates with the study to produce a tailored theatre performance around asthma knowledge and attitudes. The authors would furthermore like to thank all other members of the Achieving Control of Asthma in Children in Africa (ACACIA) steering group and coapplicants to the grant for their valuable support of the study, including Professor Chris Griffiths, Professor Robert Walton and Dr Liz Steed from Queen Mary University of London, as well as Professor Kevin Mortimer from the Liverpool School of Tropical Medicine and Professor Mike Roberts from University College London Partners.

**Contributors** GM is the main author of this manuscript and the central coordinator for ACACIA. JG is the director of the ACACIA study providing overall direction and supervision. JG has contributed directly to this manuscript by proofreading and commenting on its content. VO is a UK-based PhD student and respiratory physiologist of the study. She directly contributed to the manuscript's sections about lung function examinations and data analysis. All principal investigators (PIs) of each ACACIA centre (EA-Y, OOA, RM, HM, RN, SR and IT) have furthermore read and commented on the manuscript's content.

**Funding** This research was funded by the National Institute for Health Research (NIHR) (project reference) using UK aid from the UK Government to support global health research.

**Disclaimer** The views expressed in this publication are those of the author(s) and not necessarily those of the NIHR or the UK Department of Health and Social Care (17/63/38).

**Competing interests** JG reports personal fees from GSK, personal fees from Vifor Pharmaceuticals, personal fees from Novartis, personal fees from BV Pharma, personal fees from AstraZeneca, outside the submitted work.

**Patient and public involvement** Patients and/or the public were involved in the design, or conduct, or reporting, or dissemination plans of this research. Refer to the Methods section for further details.

**Patient consent for publication** Not required.

**Provenance and peer review** Not commissioned; externally peer reviewed.

**ORCID iD**
Gioia Mosler http://orcid.org/0000-0002-6900-4080

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
