## [Reviewer comments · BMJ Open]

ARTICLE DETAILS

TITLE (PROVISIONAL)	Achieving Control of Asthma in Children In Africa (ACACIA). Protocol of an observational study of children's lung health in six sub-Saharan African countries.
AUTHORS	Mosler, Gioia; Oyenuga, Victoria; Addo-Yobo, Emmanuel; Adeyeye, Olayinka; Masekela, R; Mujuru, Hilda; Nantanda, Rebecca; Rylance, Sarah; Ticklay, Ismail; Grigg, Jonathan

VERSION 1 – REVIEW

REVIEWER	Mandeep Jassal Division of Pediatric Pulmonology Johns Hopkins Hospital Baltimore, USA
REVIEW RETURNED	06-Dec-2019

GENERAL COMMENTS	NOTE: The following page and line numbers were derived from the PDF file. Introduction: Key points: I would consider breaking up the 2nd paragraph. It seems lengthy and it has some disparate topics. It will likely flow better when the topics are separated into different paragraphs with perhaps more details on each topic – especially on how they are interrelated and how they can contribute to your aims. 1. Page 4, Line 13: There is an immediate discussion of children only aged 13-14 years. I assume this age group is selected based on similar age ranges used by ISAAC. But it might be best to provide some context of why you are looking at this specific age group.2. Page 4, Line 25: The following statement can be found: "Some studies, have included aetiological research providing some indications about potential underlying factors of increased asthma prevalence, such as an association to air pollution." I think you should describe other risk factors. Air pollution is definitely relevant but there are other key ones that should be mentioned (e.g., allergens, viral-induced, etc.)3. Page 4, Line 31: You mentioned a study that described "sub-optimal asthma control according to the GINA questions". It would be helpful if you described what domains the GINA questions used to describe poor asthma control (e.g., symptom control, risk factors, etc.)4. Page 4, Line 35: You used the term "non-realization of therapeutic education". To me, that is a confusing term. Is there a way to more simply state that point.
---

5. Page 4, Lines 40 – 48: It seems like the SAP trial is the premise of your work. It would be helpful to know more details about the methodology that comprised that trial.

Study Aims and Objectives :

Key points: As mentioned in the introduction section, why are you focusing on children aged 12-13? Why not children aged 6-7 since ISAAC questionnaires have been used in that age group. I assumed it was because of the SAP trial but it would be nice to see that explained more in the introduction. The same question applies to why your focus on children from urban settings. Is this because SAP was done in London?

1. Page4/5: Consider splitting up the 1st sentence.
2. Page 5, Line 5/6: Perhaps you can consider consolidating the multiple outcomes and perhaps look at barriers as a tertiary aim?
3. Page 5, Line 8-10: What is “In Control” theater from SAP? It might be helpful to hear about that in the Introduction section pertinent to SAP.

Methods and Analysis:

Key points: I would be helpful for more description on Figure 1. I would prefer a slightly more detailed version of each stage so that I can better understand Figure 1 – especially given its importance to your protocol.

1. Page 5, Line 19: Why is a breathing survey in quotes, “Breathing Survey”?
2. Page 5, Line 40-41: Why did Uganda require consent and assent together?

Preparing for data collection:

1. Page 6, Line 45: What digital platform was used and why was it selected?
2. Page 6, Lines 48-52: Far greater detail is needed about Spirometry & FeNO testing. When were the ATS and ERS guidelines published? What are you using for predicted values? Who are the manufacturer’s of the equipment? Who is doing the testing since both spirometry and FeNO are highly dependent on ability of the personnel to “coach” and individual to do the right technique and get the best effort? Are you calibrating the equipment every day? Is the same person(s) doing the testing at each site to allow for reliability in results?

Recruitment:

1. Page 6, Line 10: Where are the African centers located? What cities? It might be nice to see on a map or more details in the text.
2. Page 6, Line 11: Breathing Survey is not listed in quotes but is earlier. Why?
3. Page 6, Line 11: Your target age is 12-14 years but the Introduction’s 2nd line refers to participants aged 13-14 years. Consider aligning the target ages throughout the paper.
4. Page 6, Line 14: Are kids missing school because of asthma? It would be helpful to know if there is any asthma-specific reason why it is difficult to capture children aged 12-14 in these sites.
5. Page 6, Line 17: We need more details on the local vs. central analysis given the possibility of recruiting children who are older. Previous to this point, I was only seeing mentions of children aged 12-14 years.

6. Page 6, Line 18: You mentioned that school selection is reflecting the country-specific proportions of children. Are you not focusing on children in urban settings? Are the school and clinical characteristics of kids in urban areas the same as rural areas?

Screening for Asthma Symptoms:

1. Page 6, Line 30: When approaching potential participants, what scripts or materials are you using to describe your study and what is the general contents?
2. Page 6, Line 34: How do you go about getting the parents consent – when and where?
3. Page 6, Line 35: It would be helpful to get more descriptions of the GAN questionnaire. Are you using all of the questions or some of the questions? What happens if the kids don't understand the questions? Who is going to explain it to them? What if there is underlying reading or comprehension issues in kids?

ACACIA survey and lung function tests

Key point: Can you add the ACACIA survey online as an appendix? It would be very helpful for other researchers to see.

1. Page 6, Line 40: What percentage of children do you anticipate who will be screened positive for asthma that will move on to the 2nd stage – based on previous literature?
2. Page 6, Line 40-42: We need way more description of the spirometry and FeNO – see point #2 in the Preparing for Data Collection section.
3. Page 6, Line 46: How do you identify children who “have difficulty using the digital survey”.
4. Page 7, Line 10: You mentioned the SAP survey. It would be great to see those questions in the appendix and more descriptions about each section. For example, how do you measure medication adherence?
5. Page 7, Line 15: What is the IMPALA question set? Can we see it as well or provide more context to how it captures healthcare and environmental exposures.
6. Page 7, Line 18: You mentioned that you are capturing asthma control using ACT and GINA questions. Why both? Are you comparing the two measures? What happens if both give discrepant responses?
7. Page 7, Line 25: You need more details on how you measure and normative predictive values of spirometry and FeNO.
8. Page 7, Line 27-30: Do you follow-up these kids to see if they have accessed local healthcare resources

Focus groups:

Key points: Do you have a guide or questions that will be used in the focus groups and if so, where can we find them? Who is running the focus groups and is it the same person at each site? What software are you using to analyze the scripts? How many people are analyzing the focus group data and how do they resolve discrepancies. How long are the focus groups? Where are they held? How many people are in each focus group? How do you know the selected people in the focus group are similar to the target population as a whole?

Outcomes and analysis:

Key points: A sample size calculation is needed for the primary outcome? Also you stated the following on Page 9, Line 6-9:

	A detailed statistical analysis plan is being developed with the assistance of Statistician. Focus group discussions will be transcribed and keywords identified. Qualitative results from the focus groups will in addition inform the direction and acceptability of interventions aimed at removing barriers to poor asthma control” More details on the quantitative and qualitative data analysis plan is required. What in the quantitative results is trying to be captured to tell you about barriers. How do you know when you reach thematic saturation. How do you resolve differences/discrepancies in researchers’ interpretations of the data?  1. Page 7, Line 51-53: If using the ACT, how do you address question #4 that asks about rescue inhaler or nebulizer usage. If they had not been diagnosed before, then does everyone get a 5 for that one? 2. Page 8, Line 3-23: Are you comparing the ACT scores to the other asthma control instruments (GINA...) 3. Page 8 , Line 9: Are you accounting for children who might be taking medications that affect FeNO (e.g, inhaled corticosteroids)? How are you addressing indeterminate FeNO levels (20-35 ppm)? What if kids can’t do the test? How are you coaching them to do it? 4. Page 8, Line 11-22: I would have liked to see a lot of this material earlier when describing spirometry and FeNO. How are you doing bronchodilators – MDIs or nebulizer? What dosage of bronchodilator? Who is reading the spirometry? Is it the same person or many different people since that will affect the reliability of the results. 5. Page 8, Line 26: A secondary outcome is “Percentage of children with asthma symptoms or asthma diagnosis, based on questions from the GAN screening tool”. But on Page 7, Line 48 you said the primary outcome is “the identification of the proportion of children with asthma symptoms and...” Is it a primary or secondary outcome? 6. Page 8, Lines 28-43: What are the questions/instruments being used to assess “current treatment of asthma”, “adherence to medication”, “access to medical care”, “understanding of asthma”. 7. Page 8, Line 44-46: What are the questions/instruments being used to assess Asthma-related time off from school. Who is answering this – the parents or kids? 8. Page 8, Line 46-52: What are the questions/instruments being used to assess smoking and environmental factors. Who is answering this – the parents or kids? Intervention development Key points: You stated the following on page 9, Line 17 – 21: “The ACACIA study also aims to improve the digital resources available for asthma related services and for children who suffer from asthma. Digital solutions aimed at improved asthma self-management, such as smart peak flow meters, will be tested in the context of sub-Saharan African countries.” More details are needed in the paper to describe the digital resources. What are the smart peak flow meters? Are they the only intervention being considered? If not, what others would you use? Why and how would you prioritize them? Theatre intervention development and pilot: Key point: More details are needed of the pilot study being conducted in the 5 schools.
--	--

	Page 9, Line 26: More details are the needed about the “In Control” play and why it is being used? What is the purpose or plot? Is it asthma related? You are referring the Lancet article but it will likely be better if you describe the play in more detail in the manuscript and why it is so good to translate to African environments. Page 9, Line 30: More details are required on the adaptation and validation of the play, as well as its contents. How many people do you expect to attend each play? Are all the attendees between the ages of 12-14 of will they include younger/old populations. How will you compare the effectiveness of this between countries in which you are using different theater groups to help you adapt? Page 9, Line 40: Why are only descriptive statistics being used? What are the statistics? Public and patient Involvement: I am unsure of the purpose of this section. It is logical to include comments on community involvement, but how are you measuring the impact? Who are you measuring - participants, caregivers, the general community, etc. Ethical review More details or perhaps a figure is needed to describe the ACACIA steering group. Dissemination There are no details describing how and by what means you are disseminating the study findings to the participants or communities within the study sites? Data protection and confidentiality It might be wise to describe how you are protecting identifiable data at the country-specific level
--	--

REVIEWER	Dr Justus Simba Jomo Kenyatta University of Agriculture and Technology Kenya
REVIEW RETURNED	24-Dec-2019

GENERAL COMMENTS	I am grateful to have reviewed this study protocol. Once done this study will bring a wealth of information and enhance care towards young adolescents/children with asthma in Africa. I have the following suggestions to make this study protocol clear to readers:  1. On methods and analysis page 5, make stage 4 summary clear that the gathered information will be used to adopt the UK educational theatre performance to the African set up, this becomes clear only later in the reading of the protocol. 2. On recruitment, please provide dates of when the study started or will start; make the selection of the schools clear, it should be possible for the local PIs to have an idea of the number of schools they are likely to require per selected town based on locally available data (as it is, it appears generic), specifications would make it reproducible. Further, how long is the recruitment likely to take? 3. Concerning focus groups, what will be the basis of selecting the subset of children to the focussed group discussions? 4. On outcomes and analysis, you have stated that "a detailed statistical analysis plan is being developed with assistance of statistician", it is important to have this priori data analysis plan
--

	documented including the anticipated level of measuring significance. 5. On dissemination of the results, you have stated " publications will only be made when QMUL receive written permission from the funding body.." this may suggest that you do not have full control of the data and this may limit open data sharing. Similarly, you have not declared whether data sets shall be available to other individuals outside your network. 6. Data monitoring: No specifications have been provided. Lastly, it would be good to acknowledge the potential limitation that will occur by involving different regions (in itself a good thing) which differ in culture, and perceptions towards asthma as this may have implication on the tool developed as well as the perceived control as measured by GINA and ACT. Thank you
--	--

VERSION 1 – AUTHOR RESPONSE

RESPONSES TO REVIEWER 1:

Introduction:

1. Key points: I would consider breaking up the 2nd paragraph. It seems lengthy and it has some disparate topics. It will likely flow better when the topics are separated into different paragraphs with perhaps more details on each topic – especially on how they are interrelated and how they can contribute to your aims.

a. I rearranged the section

2. 1. Page 4, Line 13: There is an immediate discussion of children only aged 13-14 years. I assume this age group is selected based on similar age ranges used by ISAAC. But it might be best to provide some context of why you are looking at this specific age group.

a. The main focus of this sentence should be on asthma symptom prevalence, I have now reworded this

b. I added further explanation of age group selection in end of introduction

3. 2. Page 4, Line 25: The following statement can be found: “Some studies, have included aetiological research providing some indications about potential underlying factors of increased asthma prevalence, such as an association to air pollution.” I think you should describe other risk factors. Air pollution is definitely relevant but there are other key ones that should be mentioned (e.g., allergens, viral-induced, etc.)

a. This paragraph focuses on what type of studies have been undertaken about childhood asthma in Africa. The main point is that little is known beyond prevalence of asthma symptoms. I have now rearranged the sentences to make this more apparent.

4. 3. Page 4, Line 31: You mentioned a study that described “sub-optimal asthma control according to the GINA questions”. It would be helpful if you described what domains the GINA questions used to describe poor asthma control (e.g., symptom control, risk factors, etc.)

a. I have added this now under ‘ACACIA survey and lung function tests’, a bit further down than suggested, where it fits well with other descriptions of tools used in the study.

5. 4. Page 4, Line 35: You used the term “non-realization of therapeutic education”. To me, that is a confusing term. Is there a way to more simply state that point.

a. Changed to: failure to carry out patient education

6. 5. Page 4, Lines 40 – 48: It seems like the SAP trial is the premise of your work. It would be helpful to know more details about the methodology that comprised that trial.

a. I have added some details about the trial, the referenced paper describes this work in full:

‘...comprising of two components: A) a theatre workshop for all children in school year groups 7 and 8, and B) a self-management workshops for children with asthma. A theory-based cluster randomised controlled trial is currently running in London to test this intervention with 3, 6, and 12 month follow-up’

Study Aims and Objectives

7. Key points: As mentioned in the introduction section, why are you focusing on children aged 12-13? Why not children aged 6-7 since ISAAC questionnaires have been used in that age group. I assumed it was because of the SAP trial but it would be nice to see that explained more in the introduction. The same question applies to why your focus on children from urban settings. Is this because SAP was done in London?

a. I have added information about age selection at the end of the introduction: ‘ACACIA will collect data related to asthma from 12 to 14 year old adolescents across sub-Saharan Africa. The study wants to understand asthma during adolescence, when there is a shift from parental to personal symptom management, and when optimal self-management standards of asthma could be introduced in a future intervention.’

b. Explanation added to text: ‘ACACIA will furthermore focus on urban areas in sub-Saharan Africa, as prevalence of childhood asthma is particularly high in urban areas of the region.’ (see also provided reference)

8. 1. Page4/5: Consider splitting up the 1st sentence.

a. done

9. 2. Page 5, Line 5/6: Perhaps you can consider consolidating the multiple outcomes and perhaps look at barriers as a tertiary aim?

a. I agree, having them all as one aim is a bit confusing. I have added barriers now as secondary aim

10. 3. Page 5, Line 8-10: What is “In Control” theater from SAP? It might be helpful to hear about that in the Introduction section pertinent to SAP.

a. I have added a little more detail here: ‘adapted from a UK-based play about asthma, called ‘In Control’

b. And more detail further down under intervention development: ‘The original ‘In Control’ theatre was written by the Nigerian-born playwright Tunde Euba in collaboration with The Tramshed - (Greenwich and Lewisham Young People’s Theatre) GLYPT as an educational performance, aimed at changing attitudes and understanding around asthma. The play revolves around a young protagonist with asthma in a school setting. It is described in a Lancet perspective article. In a UK-based feasibility study 98.7% of 1798 young people in the audience said ‘In Control’ was enjoyable and 84.6% said that the performance at least somewhat changed how they think or feel about asthma.’ (references given)

Methods and Analysis:

11. Key points: I would be helpful for more description on Figure 1. I would prefer a slightly more detailed version of each stage so that I can better understand Figure 1 – especially given its importance to your protocol.

a. more information was added

12. 1. Page 5, Line 19: Why is a breathing survey in quotes, “Breathing Survey”?

a. I have changed that

13. 2. Page 5, Line 40-41: Why did Uganda require consent and assent together?

a. The ethical review processes vary between the six countries, including different preferences about consent. We decided therefore that it was necessary to allow our partner centres in Africa the freedom to design the consent process in a manner that would be in agreement with their local ethics boards. This resulted in Uganda seeking opt-in parental consent at a different stage of the study, compared to other study centres.

Preparing for data collection:

14. 1. Page 6, Line 45: What digital platform was used and why was it selected?

a. description was added to the text: 'The digital platform comprises of a central research database, hosted by Microsoft's Azure Platform in the UK, which receives and stores the anonymised data collected by all field teams, and local databases in each centre, stored on servers within the university, which allow storage of personal data. A suite of bespoke software enables for data transfer and reporting.'

b. The decision to use Microsoft's platform and server software was made on the basis of its functionality, resilience, and security capabilities. The ACACIA study will collect in the region of 1 Million data points for analysis and has a number of unique requirements that are not supported by existing commercial products. The need to bring data together from a number of collection points and match it with both FeNO and Spirometry data also required significant bespoke development of the database.

15. 2. Page 6, Lines 48-52: Far greater detail is needed about Spirometry & FeNO testing. When were the ATS and ERS guidelines published? What are you using for predicted values? Who are the manufacturer's of the equipment? Who is doing the testing since both spirometry and FeNO are highly dependent on ability of the personnel to "coach" and individual to do the right technique and get the best effort? Are you calibrating the equipment every day? Is the same person(s) doing the testing at each site to allow for reliability in results?

a. Additional information was included in the text under Preparing for data collection, and under ACACIA survey and lung function tests

Recruitment:

16. 1. Page 6, Line 10: Where are the African centers located? What cities? It might be nice to see on a map or more details in the text.

a. I believe you mean page 7 for this item, and all following items the page number is one higher than stated in the comment

b. I have added a paragraph about the local urban centres at the end of 'Methods and Analysis': 'The study will be conducted simultaneously in schools of six African urban areas: Kumasi - Ghana; Blantyre - Malawi; Lagos – Nigeria; Durban - South Africa; Makerere – Uganda; and Harare – Zimbabwe. Work in each of these urban areas is led by a principle investigator at a local university and conducted by their team of researchers and fieldworkers.'

17. 2. Page 6, Line 11: Breathing Survey is not listed in quotes but is earlier. Why?

a. I have changed it to no quotes above

18. 3. Page 6, Line 11: Your target age is 12-14 years but the Introduction's 2nd line refers to participants aged 13-14 years. Consider aligning the target ages throughout the paper.

a. The introduction sentence refers to previous surveys, not to the ACACIA study. I have reworded this to improve understanding, and added a sentence about age selection at the end of the introduction section.

19. 4. Page 6, Line 14: Are kids missing school because of asthma? It would be helpful to know if there is any asthma-specific reason why it is difficult to capture children aged 12-14 in these sites.

a. Absenteeism is mentioned in the text as one of the reasons for children repeating school years. In most of the countries we work with students can only progress to the next school year if they pass a test, resulting in many repeated years. This results in school year groups of very mixed ages. Reasons for not passing the test to progress are varied, but include high absenteeism.

b. The question if children with asthma are left behind because of increased absenteeism is indeed interesting. Yet it is very difficult to retrieve this information. Even in the UK, schools would usually record only if absence is authorised or unauthorised, some record if absence is due to illness, but not which one. Additionally, for our six African study centres our local collaborators report parents do not commonly register with a school if their child has asthma, and it is unusual to have a school nurse or other trained personnel who could reliably report on symptoms. Given these difficulties it was decided that data collection about school absenteeism due to asthma is beyond the scope of this study.

20. 5. Page 6, Line 17: We need more details on the local vs. central analysis given the possibility of recruiting children who are older. Previous to this point, I was only seeing mentions of children aged 12-14 years.

a. The main analysis will be focusing on 12 – 14 year olds. Due to the mixed school year groups, the study centres are allowed to include additional older children in the data collection if this makes it easier to get access to children in schools. This data will be excluded for central analysis.

b. I have reorganised the paragraph a little to improve understanding

21. 6. Page 6, Line 18: You mentioned that school selection is reflecting the country-specific proportions of children. Are you not focusing on children in urban settings? Are the school and clinical characteristics of kids in urban areas the same as rural areas?

a. We recognise there may be differences in regards to different types of schools between different areas (e.g. urban and rural) of the study countries. Yet, reliable information on school types is often not available at city level.

Screening for Asthma Symptoms:

22. 1. Page 6, Line 30: When approaching potential participants, what scripts or materials are you using to describe your study and what is the general contents?

a. I added: 'The research teams are working closely with the schools to obtain consent from the parents ahead of data collection. Information sheets about the study are provided to parents and children, outlining the research, the participant's involvement, and important aspects of participation, such as the right to withdraw. The local research teams are available for any further questions and for assistance with reading of the information sheet, if necessary.'

23. 2. Page 6, Line 34: How do you go about getting the parents consent – when and where?

a. I added some additional explanation in the text. The researchers work with the schools to send out information sheets to the parents. Where required (e.g. if parents need help to understand the information), researchers would meet some of the parents as well to clarify any outstanding questions.

24. 3. Page 6, Line 35: It would be helpful to get more descriptions of the GAN questionnaire. Are you using all of the questions or some of the questions? What happens if the kids don't understand the questions? Who is going to explain it to them? What if there is underlying reading or comprehension issues in kids?

a. I added the relevant sections in the text: '... using its demographic questions, and questions 2 – 7 about breathing'

b. Researchers give an introduction to the questionnaire at the beginning of their visit in school, and are with the children while they fill in the survey. School children of 12 to 14 years of age would

usually be expected to be able to answer these questions. If any difficulties arise, the researchers would assist them with comprehension.

25. ACACIA survey and lung function tests

26. Key point: Can you add the ACACIA survey online as an appendix? It would be very helpful for other researchers to see.

a. Yes, I have added it as supplementary material

27. 1. Page 6, Line 40: What percentage of children do you anticipate who will be screened positive for asthma that will move on to the 2nd stage – based on previous literature?

a. Added section in text: existing literature suggests a prevalence of asthma symptoms between 10% and 20% in the region (reference given)

28. 2. Page 6, Line 40-42: We need way more description of the spirometry and FeNO – see point #2 in the Preparing for Data Collection section.

a. Further explanations have been added to the text

29. 3. Page 6, Line 46: How do you identify children who “have difficulty using the digital survey”.

a. The digital survey will be filled in on tablets at school in very small groups, with researchers always present to monitor their progress and assist, if necessary.

30. 4. Page 7, Line 10: You mentioned the SAP survey. It would be great to see those questions in the appendix and more descriptions about each section. For example, how do you measure medication adherence?

a. I have added the ACACIA questionnaire in the appendix, the SAP questions have been published (see reference)

31. 5. Page 7, Line 15: What is the IMPALA question set? Can we see it as well or provide more context to how it captures healthcare and environmental exposures.

a. I have reworded this. Impala encompasses a range of projects around lung health in Africa. The link provided in the references leads to the set of questionnaires used by Impala.

32. 6. Page 7, Line 18: You mentioned that you are capturing asthma control using ACT and GINA questions. Why both? Are you comparing the two measures? What happens if both give discrepant responses?

a. The predominant reason to add both tools was in order to allow for comparisons to existing studies who used either GINA or ACT. In case discrepancies between responses to GINA and ACT arise, these would be reported and discussed as part of the results analysis.

33. 7. Page 7, Line 25: You need more details on how you measure and normative predictive values of spirometry and FeNO.

a. Further explanations have been added to the text

34. 8. Page 7, Line 27-30: Do you follow-up these kids to see if they have accessed local healthcare resources

a. This would be interesting, yet it is outside the current scope of our study, and our ethical approvals do not include access to data held by the local healthcare systems.

Focus groups:

35. Key points: Do you have a guide or questions that will be used in the focus groups and if so, where can we find them? Who is running the focus groups and is it the same person at each site?

What software are you using to analyze the scripts? How many people are analyzing the focus group data and how do they resolve discrepancies. How long are the focus groups? Where are they held? How many people are in each focus group? How do you know the selected people in the focus group are similar to the target population as a whole?

a. I added further details in the focus group section and to the end of the analysis section
o I have changed focus groups with 'other community' to 'teachers' as this was a recent decision agreed upon between the study's principle investigators

Outcomes and analysis:

36. Key points: A sample size calculation is needed for the primary outcome? Also you stated the following on Page 9, Line 6-9: A detailed statistical analysis plan is being developed with the assistance of Statistician. Focus group discussions will be transcribed and keywords identified. Qualitative results from the focus groups will in addition inform the direction and acceptability of interventions aimed at removing barriers to poor asthma control”

More details on the quantitative and qualitative data analysis plan is required. What in the quantitative results is trying to be captured to tell you about barriers. How do you know when you reach thematic saturation. How do you resolve differences/discrepancies in researchers' interpretations of the data?

a. Further details added in the text

37. 1. Page 7, Line 51-53: If using the ACT, how do you address question #4 that asks about rescue inhaler or nebulizer usage. If they had not been diagnosed before, then does everyone get a 5 for that one?

a. We expect some participants not to have a rescue inhaler, alternative ways to assessing asthma control will therefore be explored as part of the analysis. Reporting of the ACT would focus either on answers to individual questions, or the ACT score would be reported for children with rescue inhalers

38. 2. Page 8, Line 3-23: Are you comparing the ACT scores to the other asthma control instruments (GINA...)

a. The main reason to use both ACT and GINA is to compare the results with other studies which used one or the other (see also answer above, page 7 (actually 8) line 18)

39. 3. Page 8 , Line 9: Are you accounting for children who might be taking medications that affect FeNO (e.g, inhaled corticosteroids)? How are you addressing indeterminate FeNO levels (20-35 ppm)? What if kids can't do the test? How are you coaching them to do it?

a. Further details have been added to the text: 'Case report forms will be used to gather information on any medications a participant is taking and the presence of any absolute or relative contraindications to FeNO or Spirometry measurement. '

b. Details on FeNO levels have been added the text: The ATS guidelines on the Interpretation of FeNO, 2011, strongly recommended that a FeNO value of (20–35 ppb in children) should be interpreted cautiously, significant FeNO result will be deemed as a result >35 ppb (<https://www.atsjournals.org/doi/full/10.1164/rccm.9120-11ST>). Testing will be performed by trained field workers according to guidelines and provided standard operating procedures.

40. 4. Page 8, Line 11-22: I would have liked to see a lot of this material earlier when describing spirometry and FeNO. How are you doing bronchodilators – MDIs or nebulizer? What dosage of bronchodilator? Who is reading the spirometry? Is it the same person or many different people since that will affect the reliability of the results.

a. Information has now been added earlier in the text, especially under ACACIA survey and lung function tests

41. 5. Page 8, Line 26: A secondary outcome is “Percentage of children with asthma symptoms or asthma diagnosis, based on questions from the GAN screening tool”. But on Page 7, Line 48 you said

the primary outcome is “the identification of the proportion of children with asthma symptoms and...”
Is it a primary or secondary outcome?

a. GAN section is moved to primary outcomes now.

42. 6. Page 8, Lines 28-43: What are the questions/instruments being used to assess “current treatment of asthma”, “adherence to medication”, “access to medical care”, “understanding of asthma”.

a. The questionnaire is now added as a supplement

43. 7. Page 8, Line 44-46: What are the questions/instruments being used to assess Asthma-related time off from school. Who is answering this – the parents or kids?

a. This is self-reported (children), see questionnaire

44. 8. Page 8, Line 46-52: What are the questions/instruments being used to assess smoking and environmental factors. Who is answering this – the parents or kids?

a. All questionnaire questions are answered by the children in school

Intervention development

45. Key points: You stated the following on page 9, Line 17 – 21:

“The ACACIA study also aims to improve the digital resources available for asthma related services and for children who suffer from asthma. Digital solutions aimed at improved asthma self-management, such as smart peak flow meters, will be tested in the context of sub-Saharan African countries.”

More details are needed in the paper to describe the digital resources. What are the smart peak flow meters? Are they the only intervention being considered? If not, what others would you use? Why and how would you prioritize them?

a. Further explanation added in the text: Digital solutions aimed at improved asthma self-management, such as smart peak flow meters and wheeze detectors, which can send information about symptoms to smart phone apps, will be tested by some of the study participants. Transferability of data infrastructure built for ACACIA to other studies or applications will also be explored.

b. Any digital resources considered to be explored should work towards the overall aims and objectives of the ACACIA study, and should only require minimal additional work for the research teams.

46. Theatre intervention development and pilot:

Key point: More details are needed of the pilot study being conducted in the 5 schools.

a. Additions were made in the text: ‘A pilot study will explore its potential to change understanding of asthma. At each study centre, the pilot study will involve performances in at a minimum of five schools with two performances in each school to audiences of up to 120 children (target age 12 – 16 yrs.), and five additional performances to local community groups, which will be defined after initial community contacts during project recruitment.’

47. Page 9, Line 26: More details are the needed about the “In Control” play and why it is being used? What is the purpose or plot? Is it asthma related? You are referring the Lancet article but it will likely be better if you describe the play in more detail in the manuscript and why it is so good to translate to African environments.

a. I have added some results from the ‘In Control’ pilot in London (with reference) and the topic area of the play. A satisfactory description of the plot would require a lot of space, but it is described in detail in the referenced Lancet article. A documentary of ‘In Control’ can be found here:

<https://myhealthinschool.org/in-control-documentary/>

48. Page 9, Line 30: More details are required on the adaptation and validation of the play, as well as its contents. How many people do you expect to attend each play? Are all the attendees between the ages of 12-14 or will they include younger/old populations. How will you compare the effectiveness of this between countries in which you are using different theater groups to help you adapt?

- a. Parts of the section are rewritten and further details added, e.g. about audience size.
- b. In regards to age: School year groups in Africa commonly have a large age range, which means that it would be difficult to limit the age of the large theatre audiences. For theatre attendance during school plays, age is therefore given as a target age, implying that older or younger children may be present during performances.
- c. Following a recent discussion within the theatre collaborators and local Principle Investigators, the target age for the child performances has now been adjusted to 12-16
- d. Yes, the theatre performances will differ between countries due to different actors and their interpretations of the play. Any between-country comparisons will remain descriptive and would be presented with reference to these potential differences.

49. Page 9, Line 40: Why are only descriptive statistics being used? What are the statistics?

- a. See above for cross-country comparisons.
- b. Answers will be primarily described in percentages of total audience (e.g. X% of the audience agreed that the performance was enjoyable), aided by visual presentations, such as box plots. Additional analysis, such as tests looking at effect significance between pre- and post-performance, might be undertaken at a later stage, e.g. if the piloted theatre intervention was considered for a larger trial.

50. Public and patient Involvement: I am unsure of the purpose of this section. It is logical to include comments on community involvement, but how are you measuring the impact? Who are you measuring - participants, caregivers, the general community, etc.

- a. This part has been rewritten

51. Ethical review

More details or perhaps a figure is needed to describe the ACACIA steering group.

- a. It is a small group of international experts, I added the name of the chair.

52. Dissemination

There are no details describing how and by what means you are disseminating the study findings to the participants or communities within the study sites?

- a. as described earlier in the text, children identified with asthma symptoms without a doctor-diagnosis of asthma, will receive a letter to the parents advising them to seek appropriate local healthcare or will be invited directly by local clinics where possible
- b. furthermore, dissemination of study results in the community is planned as part of PPIE events and activities.

53. Data protection and confidentiality

It might be wise to describe how you are protecting identifiable data at the country-specific level

- a. Addition made in text: Only a limited number of researchers would have access to the personal information, which is locally stored on servers within the partner universities.

RESPONSES TO REVIEWER 2:

I am grateful to have reviewed this study protocol. Once done this study will bring a wealth of information and enhance care towards young adolescents/children with asthma in Africa.

I have the following suggestions to make this study protocol clear to readers:

1. On methods and analysis page 5, make stage 4 summary clear that the gathered information will be used to adopt the UK educational theatre performance to the African set up, this becomes clear only later in the reading of the protocol.

o Further explanation was added in the text

2. On recruitment, please provide dates of when the study started or will start; make the selection of the schools clear, it should be possible for the local PIs to have an idea of the number of schools they are likely to require per selected town based on locally available data (as it is, it appears generic), specifications would make it reproducible. Further, how long is the recruitment likely to take?

o Recruitment time period is now added to the text

o We recognise the importance of reproducibility. However, we face several challenges to predict number of schools, and to summarise reproducible specifications of the schools selected, because:

The number of schools is not easily determined, as school sizes vary widely at our study sites (frequently between a few 100 to 1000s of students). It was furthermore not known how many students needed to be screened to reach 500 children with symptoms of asthma.

Also, a fully researcher led choice of schools is usually not feasible, as the decision to take part in the study lies with the schools

Given these difficulties, it was decided that the school selection was monitored during ongoing recruitment in regards to its representation of school type and geographical spread across the area.

3. Concerning focus groups, what will be the basis of selecting the subset of children to the focussed group discussions?

o Due to practical constraints, opportunistic sampling is used in a subset of schools

4. On outcomes and analysis, you have stated that "a detailed statistical analysis plan is being developed with assistance of statistician", it is important to have this priori data analysis plan documented including the anticipated level of measuring significance.

o Noted, yet while the analysis has been agreed, the final version of a comprehensive and detailed analysis plan is not yet available. I have therefore provided more details about the analysis in the text instead, especially in regards to focus groups and lung function results.

5. On dissemination of the results, you have stated " publications will only be made when QMUL receive written permission from the funding body.." this may suggest that you do not have full control of the data and this may limit open data sharing. Similarly, you have not declared whether data sets shall be available to other individuals outside your network.

o This arrangement is part of the current contractual agreement between QMUL and the funder. ACACIA will release curated and coded data for open access after the results have been published, within the limits of relevant data protection acts.

6. Data monitoring: No specifications have been provided.

Lastly, it would be good to acknowledge the potential limitation that will occur by involving different regions (in itself a good thing) which differ in culture, and perceptions towards asthma as this may have implication on the tool developed as well as the perceived control as measured by GINA and ACT.

Thank you

o Details on data monitoring has been added

o I agree that the recognition of cultural differences between centres will be vital during the interpretation of potential outcome differences between centres. Study results, especially from the focus groups will hopefully reveal some of the underlying cultural differences on how asthma may be perceived differently in different study centres. The protocol format however only provides little space

for discussions around potential study limitations, yet it is our firm intention to include thoughts around differences on perception of asthma in future publications on outcome differences.

VERSION 2 – REVIEW

REVIEWER	Dr. Justus Simba Jomo Kenyatta University of Agriculture and Technology, Kenya
REVIEW RETURNED	31-Jan-2020
GENERAL COMMENTS	I am satisfied with the responses given.